# Hair-follicle associated pluripotent (HAP)-cell-sheet implantation enhanced wound healing in diabetic db/db mice

Ayami Hasegawa-Haruki[1,2©], Koya Obara[2‡], Nanako Takaoka[1,2‡], Kyoumi Shirai[2‡], Yuko Hamada[2©], Nobuko Arakawa[2©], Ryoichi Aki[2‡], Robert M. Hoffman[3,4]*, Yasuyuki Amoh[1,2]*

1 Department of Dermatology, Kitasato University Graduate School of Medical Sciences, Sagamihara, Kanagawa, Japan, 2 Department of Dermatology, Kitasato University School of Medicine, Sagamihara, Kanagawa, Japan, 3 AntiCancer, Inc., San Diego, California, United states of America, 4 Department of Surgery, University of California San Diego, San Diego, California, United states of America

© These authors contributed equally to this work.
‡ KO, NT, KS and RA also contributed equally to this work.
* yasuyukiamoh@aol.com (YA); all@anticancer.com (RMH)

**Data Availability Statement:** All relevant data are within the manuscript.

**Funding:** This work was partially supported by Parents' Association Grant of Kitasato University,

## Abstract

Diabetes often results in chronic ulcers that fail to heal. Effective treatment for diabetic wounds has not been achieved, although stem-cell-treatment has shown promise. Hair-follicle-associated-pluripotent (HAP)-stem-cells from bulge area of mouse hair follicle have been shown to differentiate into keratinocytes, vascular endothelial cells, smooth muscle cells, and some other types of cells. In the present study, we developed HAP-cell-sheets to determine their effects on wound healing in type-2 diabetes mellitus (db/db) C57BL/6 mouse model. Flow cytometry analysis showed cytokeratin 15 expression in 64% of cells and macrophage expression in 3.6% of cells in HAP-cell-sheets. A scratch cell migration assay *in vitro* showed the ability of fibroblasts to migrate and proliferate was enhanced when co-cultured with HAP-cell-sheets. To investigate *in vivo* effects of the HAP-cell-sheets, they were implanted into 10 mm circular full-thickness resection wounds made on the back of db/db mice. Wound closure was facilitated in the implanted group until day 16. The thickness of epithelium and granulation tissue volume at day 7 were significantly increased by the implantation. CD68 positive area and TGF-β1 positive area were significantly increased; meanwhile, iNOS positive area was reduced at day 7 in the HAP-cell-sheets implanted group. After 21 days, CD68 positive areas in the implanted group were reduced to under the control group level, and TGF-β1 positive area had no difference between the two groups. These observations strongly suggest that the HAP-cell-sheets implantation is efficient to facilitate early macrophage activity and to suppress inflammation level. Using immuno-double-staining against CD34 and α-SMA, we found more vigorous angiogenesis in the implanted wound tissue. The present results suggest autologous HAP-cell-sheets can be used to heal refractory diabetic ulcers and have clinical promise.

School of Medicine. Ayami Hasegawa-Haruki received this award.

## Introduction

Wound healing is a very complex process which mobilizes various inflammatory cells, cytokines, chemokines, extracellular matrices, etc. to the wound site, while simultaneously increasing metabolic demand [1]. Chronic wounds are formed when healing does not proceed normally due to factors such as the underlying disease. Diabetes is one of the important causes of chronic wounds. In diabetes, more than 100 factors contribute to the wound process [2]. Diabetes affects 537 million adults worldwide [3]. Lifetime incidence of foot ulcers is likely to affect 19% to 34% of people with diabetes [4]. Diabetic foot ulcers and associated infections cause emergency department visits and hospitalization [5]. The cost of diabetic-foot-ulcer treatment is thought to exceed the cost of cancer treatment, which is an economic burden on society [6].

We previously discovered stem cells located in the bulge area of hair follicles that express nestin and are termed hair-follicle associated pluripotent (HAP) stem cells [7]. HAP stem cells from mouse and human can differentiate into neurons including dopaminergic neuron, glia, keratinocytes, smooth muscle cells, melanocytes, and beating cardiac muscle cells in vitro [8–11].

In the present study, we developed autologous HAP-cell-sheets from diabetic db/db mouse vibrissae hair follicles and demonstrated the wound-healing efficacy of the HAP-cell-sheets implantation in wounds made in the db/db mice.

## Materials and methods

### Animals and procedures

All experiments were carried out according to the guidelines of the US National Institutes of Health and were approved by the Animal Experimentation and Ethics Committees of the Kitasato University School of Medicine (Approval No. 2023–010).

A total of 31 male db/db diabetic mice (BKS.Cg-+ Leprdb/+ Leprdb/Jcl [homo]) (CLEA Japan, Tokyo, Japan) were used to prepare HAP-cell-sheets and to create a diabetic skin wound model. Six-week-old mice were weighed and kept singly in the vivarium (temperature; 24±1°C, relative humidity; 50–60%, light-dark cycle; 14 hours light—10 hours dark) for seven days prior to the experiment. The mice were randomly assigned to either the experimental (HAP-cell-sheet) or control group of the study.

### Isolation of vibrissae hair follicles from db/db mice and preparation of HAP-cell-sheets

Vibrissae hair follicles were isolated from db/db mice as described previously [9]. Fifteen mice were anesthetized with a combination of 0.75 mg/kg medetomidine, 4.0 mg/kg midazolam and 5.0 mg/kg butorphanol [12]. The upper lip skin was excised from each mouse and the vibrissae hair follicles were gently pulled out from the skin with thin tweezers. The vibrissae hair follicles were divided into three parts, those were upper, middle, and lower. As shown in Fig 1, the upper parts of the divided vibrissae hair follicles were fixed to the adhesive cell culture dish (#430165, Corning, Kennebunk, ME, USA) with 1μL of Matrigel (#356231, Corning, Bedford, MA, USA) per a hair follicle and cultured in ciKIC iPS medium (#08371–13, KANTO CHEMICAL, Tokyo, Japan) for three weeks. Then, hair follicles and the growing HAP-stem-cells were detached from the plate. The hair follicles and Matrigel were removed with tweezers and the growing HAP-stem-cells were cultured for one more week to prepare HAP-cell-sheets in DMEM (#D6429, SIGMA-ALDRICH, St Louis, MO, USA), containing 10% fetal bovine serum (FBS), 50μg/mL gentamycin (#15750–060, GIBCO, Grand Island, NY, USA), 2 mM L-glutamine (#25030149, GIBCO), and 10 mmol HEPES (#H0887,

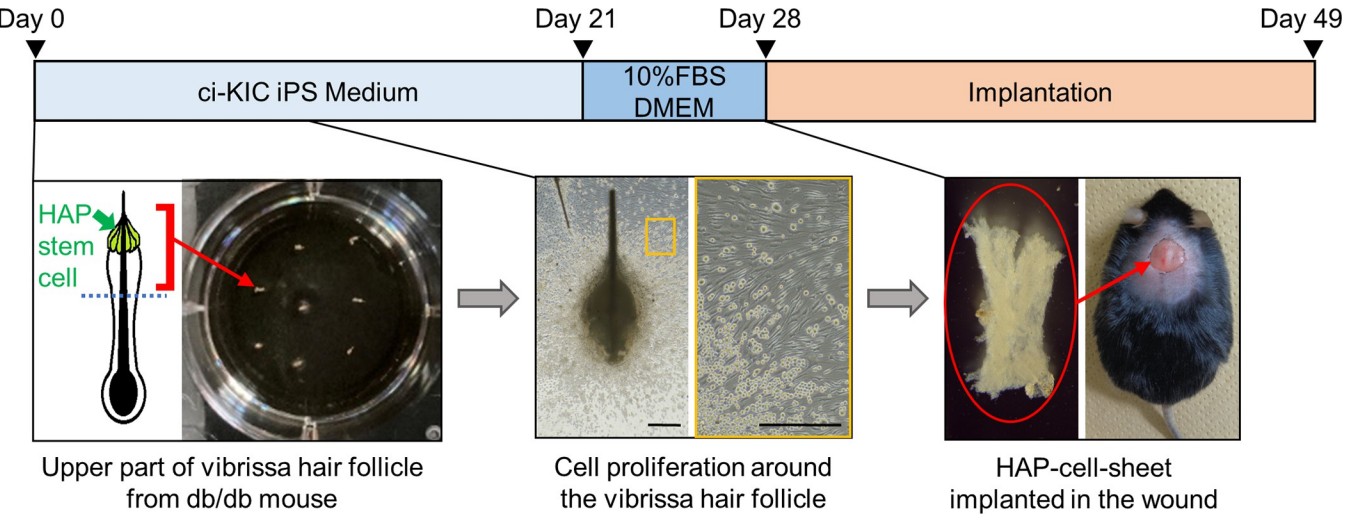

**Fig 1. Scheme for HAP-cell-sheet formation.** Scale bars = 500μm.

SIGMA-LDRICHMP). Vascular endothelial growth factor (VEGF) concentration in the supernatant of culture medium was measured after 28 days incubation, using mouse VEGF assay kit (#27102, IBL, Gunma, Japan) according to the manufacturer's instructions [13].

## Flow cytometry analysis

Flow cytometry analyses of keratin 15 (K15), CD68, CD34, CD31 and α-smooth muscle actin (α-SMA) were performed as described previously [11, 14]. The HAP-cell-sheets from the control group were used for flow cytometry analysis. For K15 and α-SMA, cells were incubated with anti-keratin 15 mouse monoclonal antibody (1:200, LHK15 #MS-1068, Lab Vision, Fermont, CA, USA) or anti-α-SMA mouse monoclonal antibody (1:200, Ab-1 #MS-113-P0, Neo-Markers, Fremont, CA, USA), then incubated with goat anti-mouse IgG H&L conjugated with phycoerythrin (PE) (1:200, ab97041, Abcam, Cambridge, UK) as the secondary antibody. Anti-mouse CD68 rabbit polyclonal antibody (1:50, #ABO10836, ABCEPTA, San Diego, CA, USA) were similarly used with goat anti-rabbit IgG H&L conjugated with PE. For CD31 and CD34, cells were incubated with anti-mouse CD31 rat monoclonal antibody (1:100, ab56299, Abcam) or anti-mouse CD34 rat monoclonal antibody (1:50, RAM34 #14-0341-82, eBioscience, San Diego, CA, USA), then with goat anti-Rat IgG conjugated with Alexa Fluor 488 (1:100, #150157, Abcam).The cells were identified by FACS Verse (BD Bioscience, Franklin Lakes, NJ, USA) and the obtained data were analyzed by FACS suite™ software (BD Bioscience).

## Scratch cell migration assay

Scratch cell migration assays were performed as reported in a previous study [15]. Mouse fibroblasts (3T3) ($1\times10^5$ cells) were seeded into each well of a 24 well cell culture plate (#353504, FALCON, Corning, NY, USA) with 10% FBS-DMEM. After 24 hours of culture to allow cells to adhere and spread on the substrate, confluent monolayer cells were scratched using a 200 μL pipette tip and washed out with PBS. Fibroblasts in the cell culture plate were then incubated with the cell culture insert (membrane 4.0μm pores, #353495, FALCON, Durham, NC, USA) containing the HAP-cell-sheet ($5\times10^5$ cells) in culture medium (experimental wells) or medium alone (control wells). After day 1, day 2 and day 3 incubation in 5%

FBS-DMEM, the cell gaps in the cell culture plate were microphotographed (BZ-X700, KEY-ENCE, Osaka, Japan). The area of the cell gaps lacking cells was measured using ImageJ software (version 1.53, National Institutes of Health, USA) [16]. VEGF concentration in the culture medium supernatant was also measured after 3 days incubation.

## db/db mouse wound-healing model

Diabetic db/db mice were anesthetized in the same way as hair follicles isolation and the pelage hair of the cranial dorsal area were shaved off. Blood sample was obtained from the tail of each mouse, and the blood glucose level was measured using a glucose meter Acc-chek (#000485, Roshe Japan, Tokyo, Japan).

Circular skin with a diameter of 10 mm was excised from the center of the shaved dorsum to inflict an excisional wound on each mouse. To prevent outflow of implanted HAP-cell-sheet, a hydrocolloid dressing (#CGF1010, Convatec, Berkshire, UK) cut into donut-shaped was fixed to the wound edge using 6–0 nylon suture (#00109190–77, Akiyama medical, Tokyo, JAPAN).

HAP-cell sheets were detached from the well by cell scraper (#90020, SPL, Gyeonggi, Korea) and the fragments of the cell sheets, containing approximate $5 \times 10^5$ cells in 100 μL of PBS were autologously implanted in the dorsal wound of each experimental-group mouse. The mice of the control group received 100 μL PBS alone in the dorsal wound. The wound was covered with a semi-occlusive adhesive dressing (PREM-RPLL, Nitto Denko, Osaka, Japan) for protection. The hydrocolloid dressing and semi-occlusive adhesive dressing were removed two days after the implant. The wounds were documented in photographs every day for 7 (9 mice) or 16 (19 mice) successive days. The wound sizes were measured by planimetric methods using ImageJ.

## Cytological staining

For cytological observation of the cultured cells, the hair follicles and growing cells of the control group were processed for immunofluorescence after three weeks of culture. Specimens were fixed with 4% paraformaldehyde and incubated with anti-CD34 rat monoclonal antibody (1:500, RAM34 #14-0341-82, eBioscience,) or CD31 rat monoclonal antibody (1:200, ab56299, Abcam), goat anti-rat IgG conjugated with Alexa Fluor 568 (1:400, ab175476, Abcam). For nuclear staining, 4',6-diamino-2-phenylindole, dihydrochloride (DAPI) (1:500, #SE196, DOJINDO, Kumamoto, Japan) was used.

## Histological staining

To observe the under-healing skin histologically, nine animals were sacrificed by euthanasia using carbon dioxide ($CO_2$) inhalation on the 7th day after implantation. The other nineteen mice were sacrificed similarly on the 21st day for healed skin observation. The wound and surrounding skin on the dorsum of the mice were excised and fixed with 4% paraformaldehyde to prepare paraffin-embedded-blocks (FFPB). The FFPB were sliced with 4 μm thickness, and the sections were stained by hematoxylin and eosin (H&E), Masson's trichrome, or immunostaining.

To identify the newly proliferated epidermis of day 7 specimen, keratin 14 (K14) in the epithelial cells was stained immunohistochemically. The FFPB sections were heat treated with boiling 10 mM sodium citrate pH 6.0 for antigen retrieval, then incubated with the first antibody anti-K14 mouse monoclonal antibody (1:100, Ab-1 #MS-115-P0, NeoMarkers). Then they were treated with the secondary-antibody, Dako REAL EnVisin HRP rabbit/mouse

(#K5007, Dako, Tokyo, Japan). The sections were developed with Dako REAL 3,3'-diamino-benzidine tetrachloride (DAB) (#5007, Dako) and then stained with Mayer's hematoxylin.

Immunofluorescent histochemistry for CD34 and α-SMA was used to evaluate angiogenesis. For fluorescence double immunostaining of CD34 and α-SMA, the FFPB sections were heat treated with boiling 1 mM EDTA pH 9.0 for antigen retrieval, then incubated with anti-CD34 rat monoclonal antibody (1:100, RAM34 #14-0341-82, eBioscience), and anti-α-SMA mouse monoclonal antibody (1:200, Ab-1 #MS-113-P0, NeoMarkers). After the first-antibody treatment, the sections were incubated with goat anti-rat IgG conjugated with Alexa Flour 488 (1:400, #A11001, Molecular Probes, Eugene, Oregon, USA) and goat anti-mouse IgG conjugated with Alexa Fluor 568 (1:400, ab175476, Abcam), and DAPI (1:500, #SE196, DOJINDO).

The levels of inflammation in the wound dermis were evaluated by expression of CD68, transforming growth factor-β1 (TGF-β1) and inducible nitric oxide synthase (iNOS). For histochemical immunostaining, the FFPB sections were heat treated with boiling 10 mM sodium citrate pH 6.0 for CD68 antigen, or 1 mM EDTA pH 9.0 for TGF-β1 antigen and iNOS antigen retrieval, then incubated with the first antibody, that is anti-CD68 rabbit polyclonal antibody (1:100, #ABO10836, ABCEPTA), anti-TGF-β1 rabbit polyclonal IgG (1:100, #sc-146, Santa Cruz, Santa Cruz, CA, USA), or anti-iNOS rabbit polyclonal antibody (1:200, #6825–4009, Biogenesis, England, UK). Secondary-antibody treatment and nuclear staining was done in the same way to K14 staining. Each staining was performed in the experimental group and in the control group.

## Microscopic observation and analysis

The H&E, Masson's trichrome and DAB-immuno-histochemical stained sections were observed with light microscope (BX51 microscope, OLYMPUS, Tokyo, Japan) and the images were captured by a digital microscopic camera (WRAYCAM-NOA2000, Wraymer, Osaka, Japan) using MicroStudio software (Wraymer). Skin thickness after wound closure in db/db mice was determined by histological section analysis. Epidermal thickness was measured in H&E stein sections and dermal thickness on Masson's trichrome stein sections. CD68, iNOS and TGF-β1 positive cell populations were measured using DAB-immunostained sections. In these measurement, two high power fields (HPF) for the 7th day and three HPFs for the 21st day in one section per a mouse were randomly selected, measured and averaged. The fluorescence immune-stained sections for CD34- and α-SMA-positive cells were observed and captured with a confocal microscopy system (LSM 710 microscope, Carl Zeiss, Oberkochen, Germany) using ZEN software. In this analysis, two HPFs in one section per a mouse were randomly selected, quantified and averaged. All quantitative analyses were performed using ImageJ software. The threshold levels were maintained constant for each analysis.

## Statistical analysis

Results are expressed as mean ± standard deviations (SD). The immunostaining analysis data at day 21 and VEGF concentration assay data were assessed after a logarithmic transformation of each variable for normalizing. Transformed variables were used only in the statistical analysis while the original values were used for presentation. Differences between groups for the histological outcomes were assessed with the Student's t-test or the Welch's t-test as appropriate. For comparison of experimental group and control mice wound healing, the wound-size data were analyzed through repeated measures ANOVA. All P values are two-tailed. A P value of less than 0.05 was considered to indicate statistical significance. Statistical analyses were done with JMP pro 17.0.0. (622753) (SAS Institute Inc., Cary NC, USA)

## Results

### Characteristics of diabetic db/db mice

The body weight (BW) of the control group was 39.0±1.9 g at follicle isolation, while BW of the HAP-cell-sheet implant experimental group was 38.6±1.6 g. The blood glucose level of the control group was 32.6±5.8 mmol/L at sacrifice, and that of the experimental group was 27.7 ±10.0 mmol/L. There was no significant difference between these characteristics of the two groups ($p > 0.05$, Student's t-test).

### HAP-cell-sheets promoted fibroblast growth *in vitro*

Flow cytometry showed keratin 15-positive cells in 64.3±23.5% of the HAP-cell-sheets from db/db mouse vibrissae hair follicles after 28 days of culture. CD34-positive cells were found in 30.0±19.5%, CD31-positivecells were found in 10.8±7.7%, α-SMA-positive cells were found in 8.1±3.8% and CD68-positive cells were found in 3.6±0.4%, of the HAP-cell-sheets (Fig 2A).

Photomicrographs of migration and/or proliferation of 3T3 fibroblasts after scratching are shown in Fig 2B. The cell-free area of the HAP-cell-sheets treated group decreased from 100% (day 0) to 69.6±6.0 (day 1), 37.8±8.0% (day 2), and 25.8±10.3% (day3). In the control group, the decrease in cell-free area was 71.2±8.0%, 48.1±11.5%, and 44.5±11.3% in order. Migration and/or proliferation of 3T3 fibroblasts were significantly enhanced in the wound by co-culture with HAP-cell-sheets (*$p < 0.05$, repeated measures ANOVA) (Fig 2C).

### HAP-cell-sheets enhanced accumulation of VEGF in the culture medium of mouse fibroblasts

VEGF was found (6.0ng/mL) in the supernatant of vibrissae hair follicles culture medium at day 28. The concentration of VEGF in the supernatant of 3T3-fibroblast culture media was 353.5±27.7 pg/mL after 3 days of culture without HAP-cell-sheets and was 1154.2±332.8 pg/mL when co-cultured with HAP-cell-sheets. The VEGF concentration was significantly increased when fibroblasts were co-cultured with HAP-cell-sheets (**$p < 0.01$, Student's t-test) (Fig 2D).

Fig 2E show the photomicrographs of cultured follicles. Proliferated cells were seen around the vibrissa hair follicles. CD34-positive cells and CD31-positive cells were observed among these cells (Fig 2E).

### HAP-cell-sheet facilitated earlier wound closure

Wounds of the HAP-cell-sheet implanted db/db mice closed significantly faster than those of the control group (**$p < 0.01$, repeated measures ANOVA) (Fig 3A and 3B). Median healing time was 16 days for the HAP-cell-sheet implanted group and 21 days for the control group, although statistically not significant ($p > 0.05$, log-rank test).

The epidermis at day 7 in the HAP transplantation group (thickness = 104.3±6.17 μm) was significantly thicker than that in the control group (87.4±5.7 μm, **$p < 0.01$, Student's t-test) (Fig 4A and 4B), however the epithelium covered area of the wound had no significant difference in both group (34.0±30.8% vs. 28.1±5.9%, $p > 0.05$, Student's t-test) (Fig 4C). The area of newly developed collagen fibers in the Masson's trichrome stained specimens was 115.9 ±29.8 $μm^2/mm^2$ at day 7 in the HAP transplantation group, statistically wider than that in the control group (85.4±24.9 $μm^2/mm^2$, *$p < 0.05$, Student's t-test) (Fig 4D and 4E). The thickness of the epidermis and dermis at day 21 had no significant difference between the HAP-cell-sheet implanted group and the control group (data not shown). However, high magnification

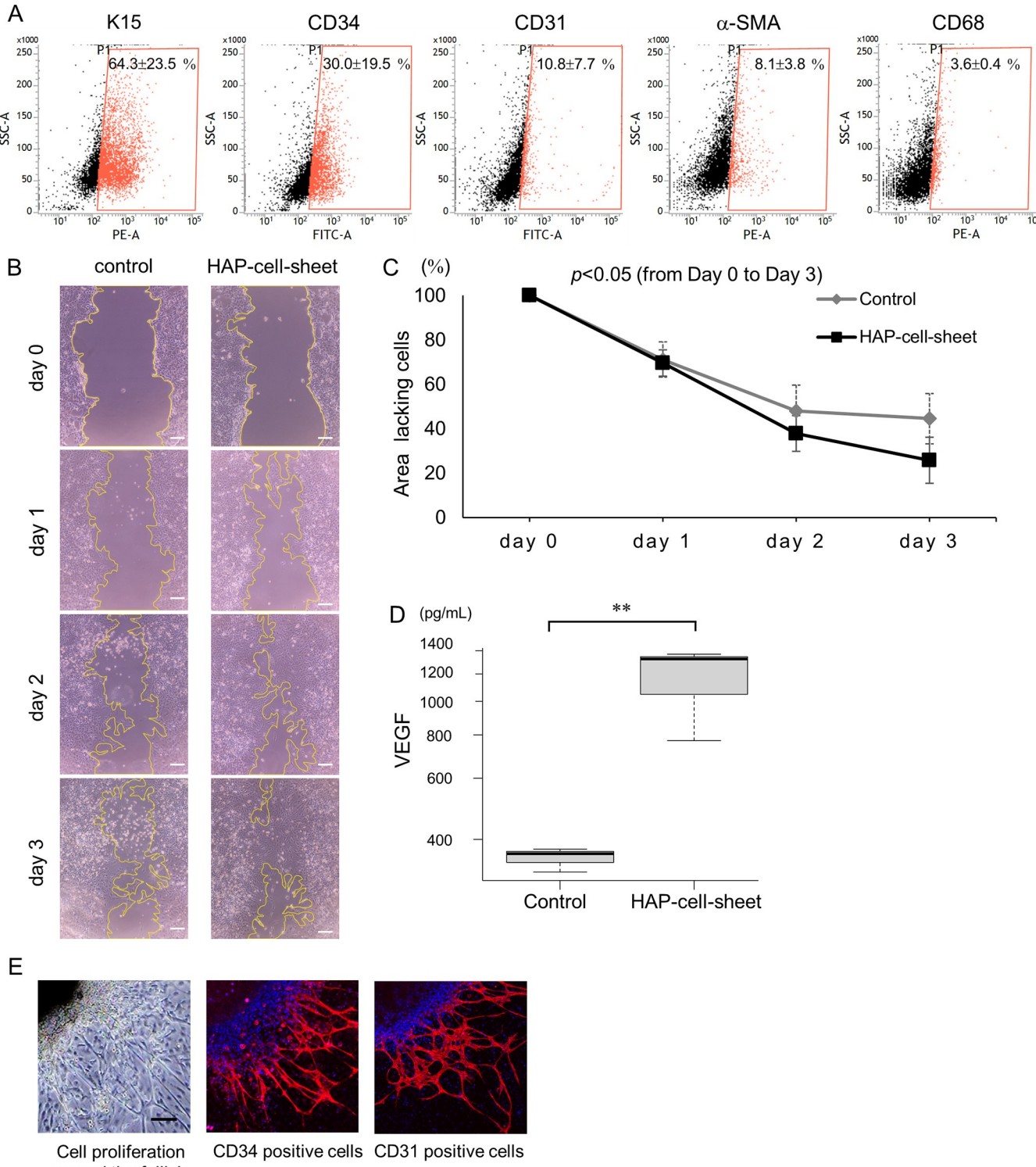

**Fig 2. HAP-cell-sheet characteristics in vitro.** (A) Flow cytometry analysis of K15-positive, CD34-positive, CD31-positive, α-SMA-positive, and CD68-positive cells in HAP-cell-sheets. (B) Analysis of HAP-cell-sheets effect on fibroblasts migration. Microscopic images of scratched 3T3 fibroblasts after 0, 1, 2 and 3 days co-culture with HAP-cell-sheets, or without (control), were displayed. The lines define lacking cell areas. Scale bars = 100 μm. (C) Time course of the cell-lacking area measured in (B). HAP-cell-sheets implantation facilitated proliferation and migration of fibroblasts (Control: n = 5, HAP-cell-sheet: n = 6. *$p < 0.05$, repeated measures ANOVA). (D) Co-culture with HAP-cell-sheets increased the concentration of VEGF in the supernatant of 3T3 fibroblast culture media (**$p < 0.01$, Student's t-test). (E) Microscopic images of proliferated cells from db/db mouse vibrissae hair follicles. CD34-positive and CD31-positive cells are distributed throughout proliferated cells. Scale bars = 100 μm.

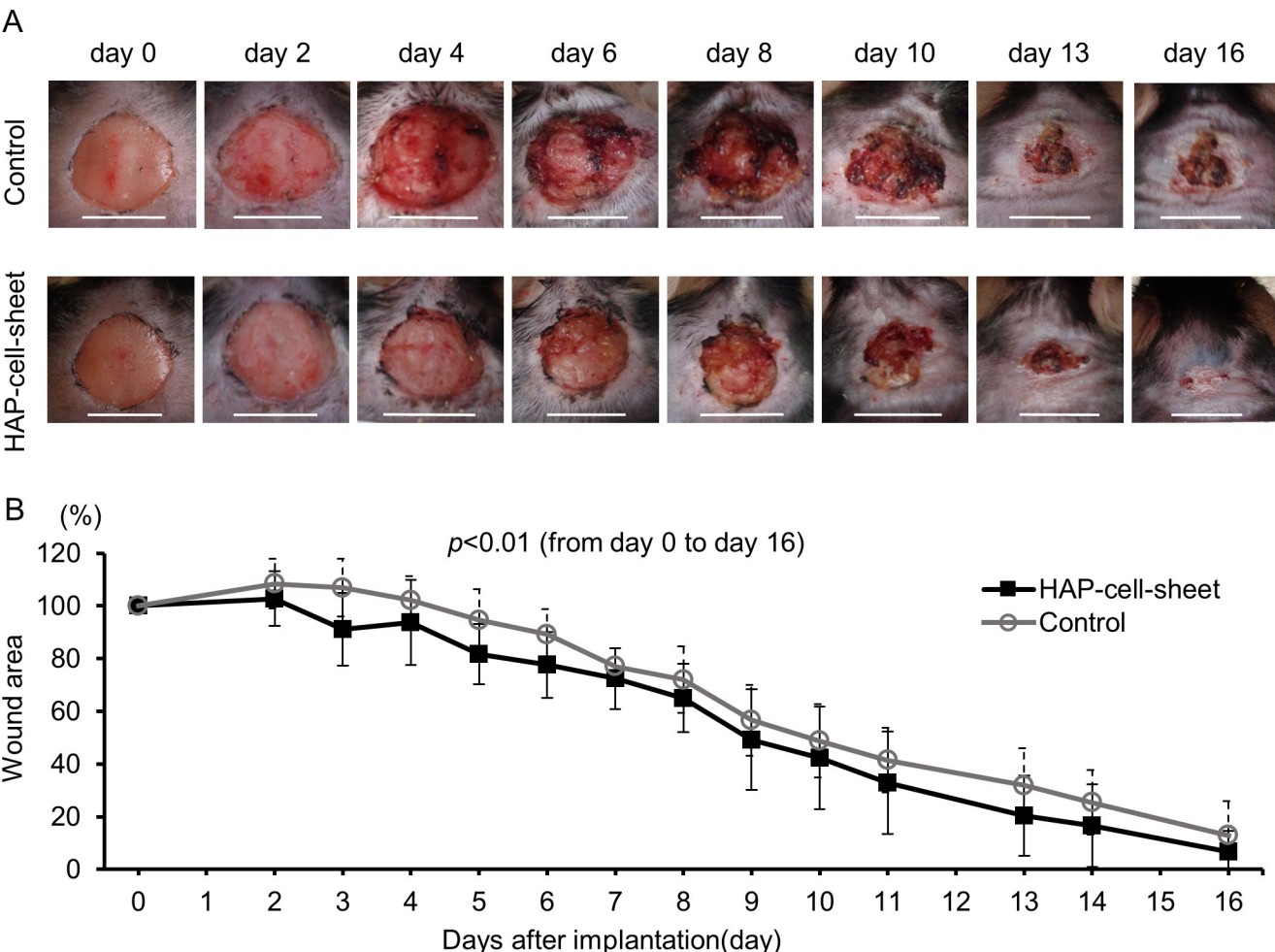

**Fig 3. Time course of wound closure of db/db mice.** (A) Photographs of closing wound for successive 16 days after wounds were made (day 0), of the HAP-cell-sheets implanted group (HAP-cell-sheet) and non-implanted control group (Control). Scale bars = 100 μm. (B) Wound area was reduced earlier in the HAP-cell-sheet implanted group when compared to that of the control group (Control: n = 7, HAP-cell-sheet: n = 8. **$p < 0.01$, multiple measures ANOVA).

microscopic observation of re-epithelialized healed wound tissue at day 21 revealed closely arranged epithelial cells in stratum basale in HAP-cell-sheet implanted group (Fig 4F).

## HAP-cell-sheets suppressed inflammation in the wound dermis

The level of inflammation in the wound dermis was evaluated with the populations of inflammation-associated cells, using immuno-stained specimens for CD68, TGF-β1 and iNOS (Fig 5A–5C). The positively stained areas at day 7 were 6,262±1,499 μm$^2$ for CD68 (Fig 5A), 1,852 ±1,003 μm$^2$ for iNOS (Fig 5B), and 4,229±697 μm$^2$ for TGF-β1 (Fig 5C) in the HAP-cell-sheet implanted group. Those in the control group were 3,210±1447 μm$^2$ for CD68, 5,196±1,150 μm$^2$ for iNOS, and 1,837±859 μm$^2$ for TGF-β1 respectively. The expression of CD68 and TGF-β1 was significantly higher, and that of iNOS was significantly lower in the macrophages of HAP-cell-sheet implanted group than those of the control group (CD68: ***$p < 0.001$, TGF-β1: ***$p < 0.001$, iNOS: ***$p < 0.001$, Student's t-test) (Fig 5D, 5H and 5F).

The positively stained areas at day 21 were 0.20±0.20 mm$^2$ for CD68 (Fig 5A), 0.20±0.53 mm$^2$ for iNOS (Fig 5B), and 1.07±1.74 mm$^2$ for TGF-β1 (Fig 5C) in the HAP-cell-sheet

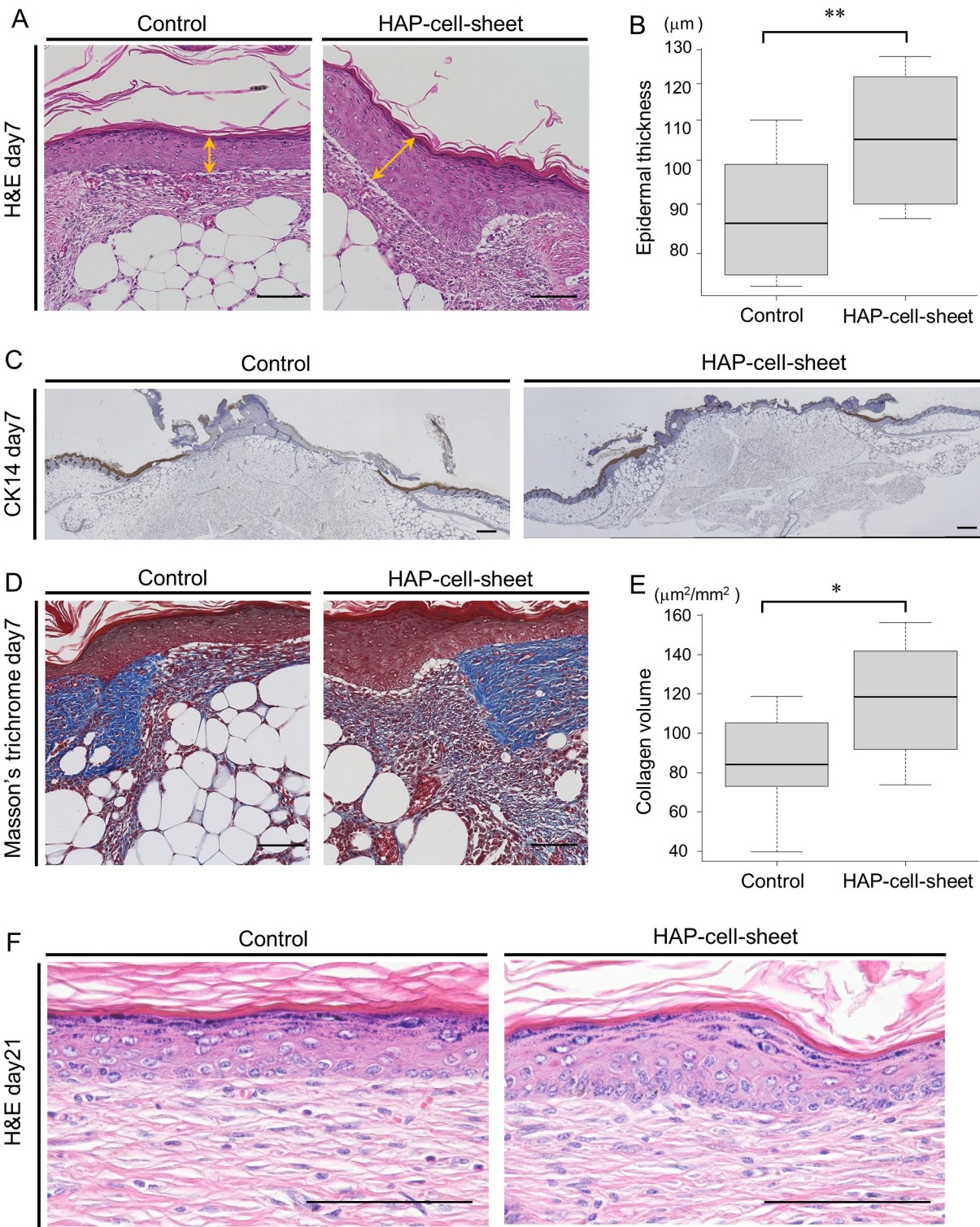

**Fig 4. Histology of wound tissues at day 7and day 21.** (A) Photomicrographs of wound tissues stained with H&E. Scale bars = 100 μm. (B) Quantitative analysis of epidermal thickness. Newly generated epidermis was thicker in the HAP-cell-sheet implanted group (Control: n = 4, HAP-cell-sheet: n = 5. **$p < 0.01$, Student's t-test). (C) Low magnification photomicrographs of wound tissues stained with anti-Keratin 14 antibody. Scale bars = 1mm. (D)Specimens of wound tissues stained with Masson's trichrome. Scale bars = 100 μm. (E) Quantitative analysis of granulation tissue. Stained collagen area was increased in the HAP-cell-sheet implanted group (Control: n = 4, HAP-cell-sheet: n = 5. *$p < 0.05$, Student's t-test). (F) High magnification photomicrographs of re-epithelialized healed wound tissue at day 21 (H&E). Epithelial cells of stratum basale in HAP-cell-sheet implanted group were arranged closely so that count larger number.

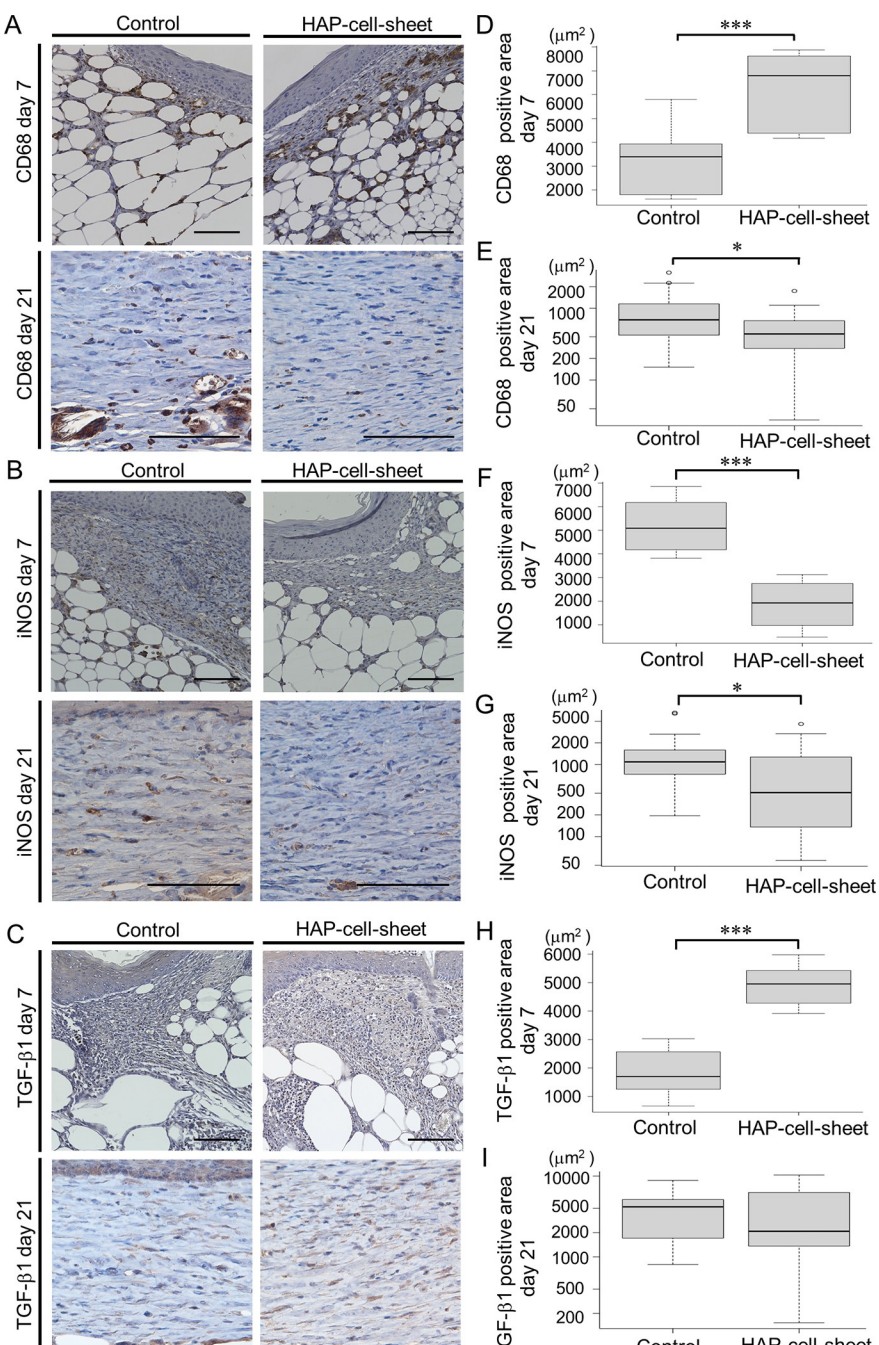

**Fig 5. Immunocytochemically stained specimens for CD68, iNOS and TGF-β1.** (A-C) Photomicrographs of wound tissues stained with anti-CD68, anti-iNOS, or anti-TGF-β1 antibodies. Scale bars = 100 μm. (D) Quantitative analysis of CD68 positive cells at day 7 (Control: n = 4, HAP-cell-sheet: n = 5. ***$p < 0.001$). (E) Quantitative analysis of CD68 positive cells at day21 (Control: n = 7, HAP-cell-sheet: n = 8. *$p < 0.05$). (F) Quantitative analysis of iNOS positive cells at day 7 (Control: n = 4, HAP-cell-sheet: n = 5. ***$p < 0.001$). (G) Quantitative analysis of iNOS positive cells at day21. (Control: n = 7, HAP-cell-sheet: n = 8. *$p < 0.05$). (H) Quantitative analysis of TGF-β1 positive cells at day 7 (Control: n = 4, HAP-cell-sheet: n = 5. ***$p < 0.001$). (I) Quantitative analysis of TGF-β1 positive cells at day 21 (Control: n = 7, HAP-cell-sheet: n = 8. $p > 0.05$, Welch's t-test). Student's t-test were used for statistical comparison except (I).

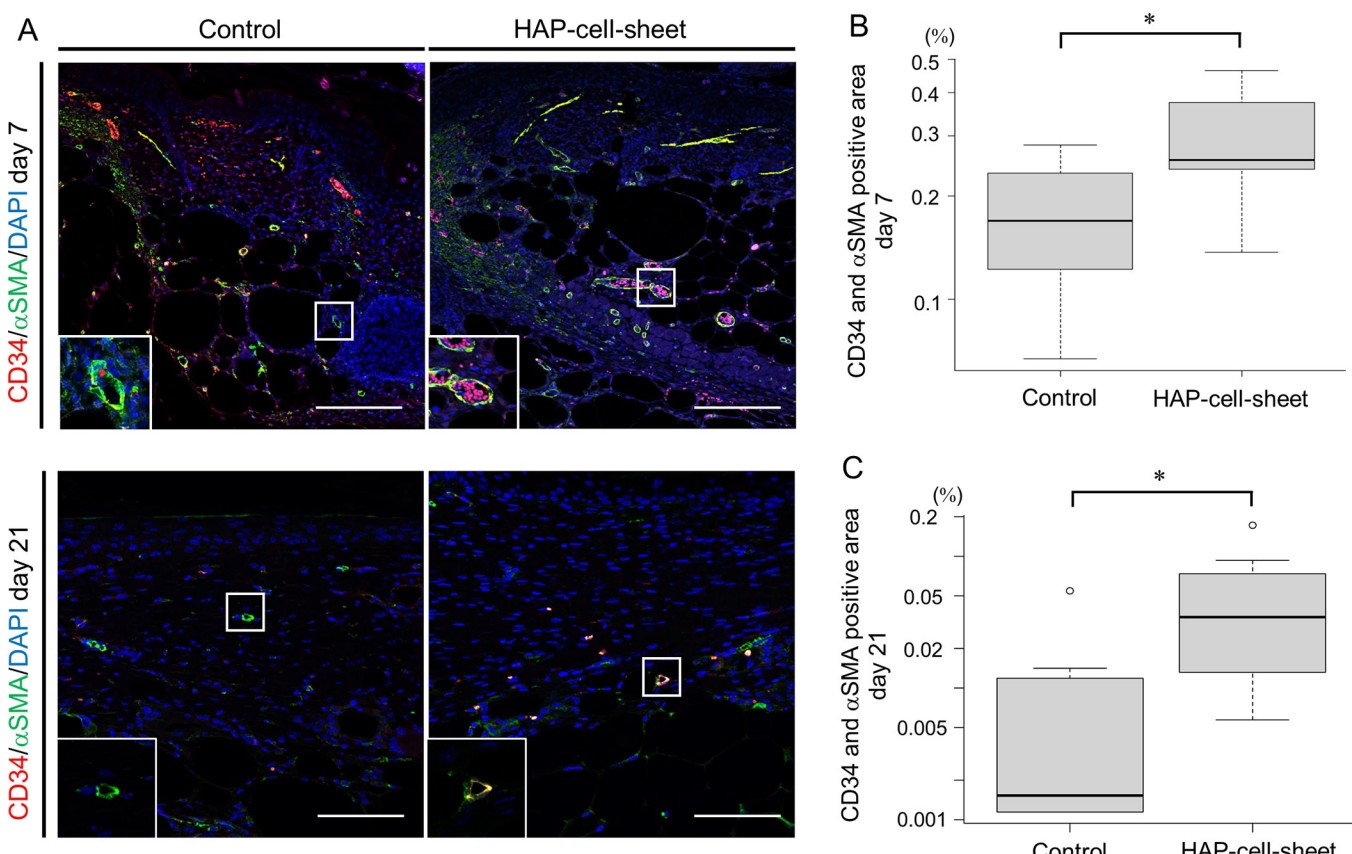

**Fig 6. Angiogenesis observed in wound tissue at day 7 and day 21.** (A) Immunofluorescence photomicrographs of CD34 (red) and α-SMA (green). Nuclei were stained by DAPI (blue). Double positive areas were expressed near yellow color. Scale bars = 100 μm. (B and C) Quantitative analysis of CD34 and α-SMA in immunofluorescence-stained specimen at day 7 (B) and 21 (C). Double positive area increased in the HAP-cell-sheet implanted group (day 7; *$p < 0.05$, Student's t-test. day 21; *$p < 0.05$, Welch's t-test).

implanted group. Those in the control group were 0.38±0.40 mm$^2$ for CD68, 0.49±0.74 mm$^2$ for iNOS, and 1.70±1.28 mm$^2$ for TGF-β1 respectively. The expression of CD68 and iNOS was significantly lower in the macrophages of HAP-cell-sheet implanted group than those of the control group (CD68: *$p < 0.05$, iNOS: *$p < 0.05$, Student's t-test) (Fig 5E and 5G). The expression of TGF-β1 had no significant difference between both groups (TGF-β1: $p > 0.05$, Student's t-test) (Fig 5I).

## Implantation of HAP-cell-sheet promoted angiogenesis in db/db mice

Newly-constituted microvessels in the wound tissues were identified as positive for CD34 and α-SMA (Fig 6A). Existing mature vessels were identified as tube structures stained against only for α-SMA. The percentage of newly-developed microvessels had significantly increased in the HAP-cell-sheet implanted mouse group (0.29±0.11%) compared to that (0. 17±0.072%) of the control group (*$p < 0.05$, Student's t-test) (Fig 6B) at day 7. Similar result was observed at day 21 (Control: 0.0040±0.021%, HAP-cell-sheet: 0.031±0.060%), (*$p < 0.05$, Student's t-test) (Fig 6C).

## Discussion

The present study demonstrate that implantation of HAP-cell-sheets promotes wound closure in diabetic db/db mice. Our FACS analyses showed 64% of the cells in HAP-cell-sheets were

keratinocytes. Keratinocyte progenitor cells expressing keratin-15 distributed in the bulge region have been shown to play an important role in epidermal regeneration in wound healing [17]. Topical application of keratinocyte sheets has been reported to be effective in the treatment of diabetic wounds in patients [18]. It is thought that HAP-cell-sheet implantation may stimulat keratinocyte migration and led to faster re-epithelialization. Implantation of HAP-cell-sheets resulted in thickening the epithelium and granulation tissue on day 7. We found no difference between the re-epithelization rate of HAP-cell sheets implanted group and that of the control group at day 7, also between the median healing time, but wound closure was facilitated in the implanted group until day 16. Neo regenerated epidermis in healed wound tissue at day 21 had similar thickness in both groups, but epithelial cells in the basal layer was closely arranged in the HAP-cell-sheet implanted mice. Thicker epithelium at day 7 in implanted group might account for the high-quality epidermis at day 21.

It has been well documented that granulation tissue composed of fibroblasts and collagen supports re-epithelialization as a scaffold for keratinocytes to proliferate and to migrate into the wound site [19, 20].

In db/db mice, it has been reported that maturation of granulation tissue in wounds was slower and collagen deposition was markedly delayed [21], and that the poor healing quality of db/db wounds was due to reduced epithelial thickness with irregular keratinocyte arrangement and significantly thinner granulation tissue [22]. Implantation of HAP-cell-sheets has a probability of improvement in wound healing by redeeming these characteristics in db/db mouse.

The responsiveness of dermal TGF-β has been shown to mediates wound contraction and epithelial closure [23]. The decreased TGF-β signaling and collagen levels in dermis has been reported to delay wound closure by affecting fibloblast proliferation and vascular cell population [23, 24]. We demonstrated that the implantation of HAP-cell-sheets to db/db mouse wound induced increase of TGF-β1 level in dermis at day 7. HAP-cell-sheets implantation may also have a possibility to help wound healing in db/db mouse by increasing dermal TGF-β1 level.

VEGF-A, which is required for angiogenesis, was identified from the culture supernatant of vibrissae hair follicles culture medium and from the supernatant of mouse fibroblasts culture medium in the present study. Accumulated VEGF-A level in the fibroblasts culture medium was significantly higher when cultured with HAP-cell-sheets. VEGF-A is reduced in diabetic mice [25], and it has been reported that treating wounds with VEGF-A restored angiogenesis [26]. The main sources of VEGF-A are keratinocytes and macrophages [27, 28], therefore, it is thought that VEGF was produced from macrophages and keratinocytes differentiated from the HAP-cell-sheets and promoted angiogenesis.

We previously reported that new blood vessels grew from implanted hair follicles [29]. In the present study, we observed CD31-positive cells among the proliferated cells around the cultured follicles to confirm that HAP-stem-cells have ability to differentiate into blood vessels. CD34 is expressed not only by hematopoietic stem cells but by vascular endothelial progenitors, fibroblasts, epithelial progenitors, and more [30]. FACS analysis in the present study showed that CD34 was present in 30% of the cells in the HAP-cell-sheets. The proliferated cells around the cultured follicles also contained CD34-positive cells. Angiogenesis promoted with implantation of vascular endothelial progenitors has been well documented by many researchers [31]. Immuno-double-staining against CD34 and a-SMA is helpful to investigate angiogenesis [32]. Newly-constituted double-positive blood vessels in the wound tissues were increased in the HAP-cell-sheet implanted mouse group at day 7 and day 21. HAP-cell-sheets, containing epithelial progenitors and endothelial progenitors, facilitate not only in re-epithelialization but also in angiogenesis.

Wounds of implanted HAP-cell-sheets were healed with low inflammation. Our flow cytometry showed 3.6% of the HAP-cell-sheet cells differentiated into macrophages. Macrophages exert phagocytic and pro-inflammatory effects in the early stages of wounds and induce re-epithelialization and angiogenesis from around three days after wounding [33]. Macrophages also produce TGF-β1, which attracts fibroblasts to the wound area [34], and it was reported that TGF-β1 were reduced in diabetic foot ulcers [35]. Exhaustion of macrophages induces the reduction of TGF-β1 and VEGF [36]. It is well documented that there is a mutual feedback regulation between iNOS and TGF-β1 [37]. Decrease of TGF-β1 may lead to increase in iNOS [38], and the production of large amounts of NO can lead to damage to surrounding tissues [39]. Macrophages supplied by HAP-cell-sheets may have been involved in reducing the inflammation in the wound.

In conclusion, the HAP-cell-sheets implantation closed the wounds faster than untreated wounds, by facilitating in epithelial cell proliferation and in maturation of granulation tissue, and by suppression of inflammation. These effects were confirmed by measuring epidermal thickness and mass of collagen, and by comparing the levels of TGF-β1, CD68 and iNOS in the wound tissue. Furthermore, the HAP-cell-sheets increased VEGF concentration in culture medium and promoted angiogenesis by implantation. The present results suggest autologous HAP-cell-sheets can be used to heal refractory diabetic ulcers and have clinical promise.

## Acknowledgments

We thank Mari Mori and Masako Ishii for cooperation of the technical assistance.

## Author Contributions

**Conceptualization:** Ayami Hasegawa-Haruki, Koya Obara, Kyoumi Shirai, Yuko Hamada, Nobuko Arakawa, Ryoichi Aki, Robert M. Hoffman, Yasuyuki Amoh.

**Data curation:** Ayami Hasegawa-Haruki.

**Formal analysis:** Ayami Hasegawa-Haruki, Yuko Hamada.

**Funding acquisition:** Ayami Hasegawa-Haruki.

**Investigation:** Ayami Hasegawa-Haruki, Koya Obara, Nanako Takaoka, Yuko Hamada, Nobuko Arakawa.

**Methodology:** Ayami Hasegawa-Haruki, Koya Obara, Yuko Hamada, Nobuko Arakawa.

**Supervision:** Robert M. Hoffman, Yasuyuki Amoh.

**Validation:** Ayami Hasegawa-Haruki, Yuko Hamada.

**Visualization:** Ayami Hasegawa-Haruki.

**Writing – original draft:** Ayami Hasegawa-Haruki.

**Writing – review & editing:** Ayami Hasegawa-Haruki, Robert M. Hoffman, Yasuyuki Amoh.

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
