## [Decision Letter · Decision Letter 0]

12 Dec 2023

PONE-D-23-39123Hair-follicle associated pluripotent (HAP)-cell-sheet implantation enhanced wound healing in diabetic db/db micePLOS ONE  Dear Dr. Hasegawa-Haruki, Thank you for submitting your manuscript to PLOS ONE. After careful consideration, we feel that it has merit but does not fully meet PLOS ONE’s publication criteria as it currently stands. This research paper, titled "Hair Follicle-Associated Pluripotent (HAP) Stem Cell Sheet Transplantation Enhances Wound Healing in Diabetic db/db Mice," explores the treatment of chronic ulcers, particularly diabetic wounds, using Hair Follicle-Associated Pluripotent (HAP) stem cells. The findings indicate that autologous HAP stem cell sheets have potential clinical value in treating refractory diabetic ulcers. I'm including feedback from two expert reviewers regarding your manuscript. Their insights highlight the study's potential while also presenting various critiques and recommendations. After read reviewer’s feedback, I believe this article requires careful revision. This article falls short in several aspects, including whether it is a significant contribution to the field, its organization and comprehensive description, its scientific soundness and accuracy, and the correctness and readability of the English used. All of these areas require a round of revisions for improvement. So, I decide give you major revision decision. It's our hope that you find this feedback constructive. Should additional data be available to address the raised concerns, we are open to examining a modified submission. Our timeline is adaptable for any necessary extended research. For submissions linked to a specific thematic issue, please liaise with the editorial team concerning resubmission deadlines. We emphasize that key findings must be backed by comprehensive statistics from ample independent trials. Manuscripts may be returned at the editor's discretion for the completion of any lacking or partial statistical data.

I strongly suggest rearranging the figure captions so they are more reader-friendly. Ideally, the captions should be placed at the end or integrated within the figures themselves. The current layout, with captions and images misaligned, is quite confusing for the reader. For more details, please refer to the layout instructions of PLOS ONE. We invite you to submit a revised version of the manuscript that addresses the points raised during the review process.

We look forward to receiving your revised manuscript.

Yours sincerely,

Jimin Han

Academic Editor,

PLOS ONE

Journal Requirements:

Please ensure that your manuscript meets PLOS ONE's style requirements, including those for file naming. The PLOS ONE style templates can be found athttps://journals.plos.org/plosone/s/file?id=wjVg/PLOSOne_formatting_sample_main_body.pdf and
https://journals.plos.org/plosone/s/file?id=ba62/PLOSOne_formatting_sample_title_authors_affiliations.pdf
We note that Figures 1 and 3 in your submission contain copyrighted images. All PLOS content is published under the Creative Commons Attribution License (CC BY 4.0), which means that the manuscript, images, and Supporting Information files will be freely available online, and any third party is permitted to access, download, copy, distribute, and use these materials in any way, even commercially, with proper attribution. For more information, see our copyright guidelines: http://journals.plos.org/plosone/s/licenses-and-copyright.

1. You may seek permission from the original copyright holder of Figure1 and 3 to publish the content specifically under the CC BY 4.0 license.

We suggest you thoroughly copyedit your manuscript for language usage, spelling, and grammar. If you do not know anyone who can help you do this, you may wish to consider employing a professional scientific editing service.

Whilst you may use any professional scientific editing service of your choice, PLOS has partnered with both American Journal Experts (AJE) and Editage to provide discounted services to PLOS authors. Both organizations have experience helping authors meet PLOS guidelines and can provide language editing, translation, manuscript formatting, and figure formatting to ensure your manuscript meets our submission guidelines. To take advantage of our partnership with AJE, visit the AJE website (http://learn.aje.com/plos/) for a 15% discount off AJE services. To take advantage of our partnership with Editage, visit the Editage website (www.editage.com) and enter referral code PLOSEDIT for a 15% discount off Editage services.If the PLOS editorial team finds any language issues in text that either AJE or Editage has edited, the service provider will re-edit the text for free.

We noticed you have some minor occurrence of overlapping text with the following previous publication(s), which needs to be addressed:

https://www.jidonline.org/article/S0022-202X(23)02306-0/fulltext

In your revision ensure you cite all your sources (including your own works), and quote or rephrase any duplicated text outside the methods section. Further consideration is dependent on these concerns being addressed.

Reviewers' comments:

Reviewer's Responses to Questions

**Comments to the Author**

1. Is the manuscript technically sound, and do the data support the conclusions?

Reviewer #1: Yes

Reviewer #2: Partly

2. Has the statistical analysis been performed appropriately and rigorously? 

Reviewer #1: Yes

Reviewer #2: Yes

3. Have the authors made all data underlying the findings in their manuscript fully available?

Reviewer #1: Yes

Reviewer #2: Yes

4. Is the manuscript presented in an intelligible fashion and written in standard English?

Reviewer #1: Yes

Reviewer #2: Yes

5. Review Comments to the Author

Reviewer #1: Major concerns

Authors constructed hair-follicle associated pluripotent (HAP)-cell-sheet for implantation to enhance wound healing in diabetic db/db mice. However, authors missed several critical factors.

1. The cell sheet contains a variety of cells, but which cell plays the critical role is unknown.

2. There was no significant difference in histological results. Moreover, the epidermis and dermis in HAP group were thicker, which was not favorable for tissue regeneration.

3. Why could CD34-positive cells represent neovascularization, does the cell sheet still express CD34 after cell differentiation?

4.What is the biological role of CD34?

Minor concerns

1. The grammar needs to be improved thoroughly over the manuscript.

Reviewer #2: This study entitled 'Hair-follicle associated pluripotent (HAP)-cell-sheet implantation enhanced wound healing in diabetic db/db mice' focuses on utilizing HAP cells derived from hair follicles for treating excisional wound in diabetic conditions. The manuscript requires further refinement before it can satisfy the standards for publication.

major:

1. Concerning the use of flow cytometry for cell characterization from the HAP sheet: a) There appears to be no use of isotype controls in the process. b) CD31 should be stained to exclude endothelial cells. c) In Figure 2A, there are discrepancies in SSC-A (cell granularity) on the y-axis across the four charts. If these cells are from the same batch, uniformity is expected. An explanation for this variation is needed.

2. Regarding the transplantation of 5 × 10^5 cells for a 1 cm circular wound, I am skeptical about the effectiveness. Has there been any comparison with other cell types, such as keratinocytes? How many cells are typically implanted? Additionally, why is this particular type of cell, which requires a month to culture, considered more effective?

3. The claim that the HAP-cell-sheet accelerates wound healing is not convincingly supported by the presented results: a) The sheet does not seem to properly align with the wound shape. b) Re-epithelialization is crucial for wound closure. Has there been an assessment of when re-epithelialization occurs in db/db mice, particularly whether it happens sooner in the HAP-sheet treated group compared to the control group? It would be beneficial to perform Krt14 staining and H&E staining around the time of re-epithelialization. c) The survival duration of the transplanted cells in relation to wound healing is uncertain. It's important to demonstrate the presence of transplanted cells throughout the healing process.

minor:

1. The structure of the article could be improved for better clarity. I suggest organizing it as follows: a) Isolation and characterization of HAP sheet cells. b) The role of HAP-cell-sheets in accelerating wound healing. c) The influence of HAP-cell-sheets on fibroblast growth and VEGF accumulation. d) The impact of HAP-cell-sheets on promoting angiogenesis. e) How HAP-cell-sheets may reduce inflammation.

2. The process of isolating HAP is not sufficiently detailed: a) What methods are employed to prevent contamination? b) Regarding lines 91-93, are the plates coated with Matrigel that detaches along with the cells? If Matrigel is part of the sheet, consider including a control group using only Matrigel sheets in both the wound scratch assay and in vivo healing studies. If there's no Matrigel, please clarify how the liquid cell pellet is maintained at the wound's center. c) Over the course of the three-week culture period, how frequently was the medium changed?

3. In Figure 2A, it would be beneficial to include the gating strategy used in the flow cytometry analysis.

4. Concerning line 251, the statement "CD34-positive cells were observed around the follicles (Fig 2B)" needs clarification. What is the intended message here?

5. In Figure 7A, please specify which section of the wound is being shown.

6. PLOS authors have the option to publish the peer review history of their article (what does this mean?). If published, this will include your full peer review and any attached files.

Reviewer #1: No

Reviewer #2: No

---

## [Author Response · Author response to Decision Letter 0]

4 Apr 2024

We would like to thank the reviewers and editors for their time and effort to review this manuscript. 

Please consider the attached manuscript that has been revised according to the reviews.

For responses to each reviewer's comments, please refer to ‘Response to Reviewers’.

We believe that this manuscript will be of great interest to your reader and we hope that you find it acceptable for publication in Plos One.

---

## [Decision Letter · Decision Letter 1]

16 May 2024

Hair-follicle associated pluripotent (HAP)-cell-sheet　implantation enhanced wound healing in diabetic db/db mice

PONE-D-23-39123R1

Dear Dr. Hasegawa-Haruki,

We’re pleased to inform you that your manuscript has been judged scientifically suitable for publication and will be formally accepted for publication once it meets all outstanding technical requirements.

Kind regards,

Jimin Han

Academic Editor

PLOS ONE

Additional Editor Comments (optional):

Article ready to accept.

Reviewers' comments:

Reviewer's Responses to Questions

**Comments to the Author**

1. If the authors have adequately addressed your comments raised in a previous round of review and you feel that this manuscript is now acceptable for publication, you may indicate that here to bypass the “Comments to the Author” section, enter your conflict of interest statement in the “Confidential to Editor” section, and submit your "Accept" recommendation.

Reviewer #3: All comments have been addressed

Reviewer #4: All comments have been addressed

2. Is the manuscript technically sound, and do the data support the conclusions?

Reviewer #3: Yes

Reviewer #4: Yes

3. Has the statistical analysis been performed appropriately and rigorously? 

Reviewer #3: Yes

Reviewer #4: Yes

4. Have the authors made all data underlying the findings in their manuscript fully available?

Reviewer #3: Yes

Reviewer #4: Yes

5. Is the manuscript presented in an intelligible fashion and written in standard English?

Reviewer #3: Yes

Reviewer #4: Yes

6. Review Comments to the Author

Reviewer #3: The manuscript has been revised to address all the errors identified in the reviewer's comments. I have carefully reviewed the changes and agree to the publication of the article.

Reviewer #4: (No Response)

7. PLOS authors have the option to publish the peer review history of their article (what does this mean?). If published, this will include your full peer review and any attached files.

Reviewer #3: No

Reviewer #4: No

---

## [Editor Report · Acceptance letter]

21 May 2024

PONE-D-23-39123R1 

PLOS ONE

Dear Dr. Hasegawa-Haruki, 

I'm pleased to inform you that your manuscript has been deemed suitable for publication in PLOS ONE. Congratulations! Your manuscript is now being handed over to our production team.

Kind regards, 

on behalf of

Dr. Jimin Han 

Academic Editor

PLOS ONE